# A New Method for Determination of Thymol and Carvacrol in *Thymi herba* by Ultraperformance Convergence Chromatography (UPC^2^)

**DOI:** 10.3390/molecules25030502

**Published:** 2020-01-23

**Authors:** Xiaoqiang Chang, Peng Sun, Yue Ma, Dongchen Han, Yifan Zhao, Yue Bai, Dong Zhang, Lan Yang

**Affiliations:** 1Institute of Chinese Materia Medica, China Academy of Chinese Medical Sciences, Beijing 100700, China; 15538186968@163.com (X.C.); psun@icmm.ac.cn (P.S.); yuema_2016@126.com (Y.M.); prefect_flight@126.com (D.H.); zyfan_666@163.com (Y.Z.); by15076380819@163.com (Y.B.); 2Artemisinin Research Center, China Academy of Chinese Medical Sciences, Beijing 100700, China

**Keywords:** *Thymus mongolicus*, *Thymus przewalskii*, thymol, carvacrol, ultraperformance convergence chromatography

## Abstract

Ultraperformance convergence chromatography is an environmentally friendly analytical technique for dramatically reducing the use of organic solvents compared to conventional chromatographic methods. In this study, a rapid and sensitive ultraperformance convergence chromatography method was firstly established for quantification of thymol and carvacrol, two positional isomers of a major bioactive in the volatile oil of *Thymi herba*, the dried leaves and flowers of *Thymus mongolicus* or *Thymus przewalskii*, known in China as “Dijiao.” Using a Trefoil^TM^ CEL1 column, thymol and carvacrol were separated in less than 2.5 min and resolution was enhanced. The method was validated with respect to precision, accuracy, and linearity according to the National Medical Products Administration guidelines. The optimized method exhibited good linear correlation (*r* = 0.9998−0.9999), excellent precision (relative standard deviations (RSDs) < 1.50%), and acceptable recoveries (87.29–102.89%). The limits of detection for thymol and carvacrol were 1.31 and 1.57 ng/L, respectively, while their corresponding limits of quantification were 2.63 and 3.14 ng/L. Finally, the quantities of the two compounds present in 16 *T. mongolicus* and four *T. przewalskii* samples were successfully evaluated by employing the developed method. It is hoped that the results of this study will serve as a guideline for the quality control of *Thymi herba*.

## 1. Introduction

*Thymi herba* is the dried leaves and flowers of *Thymus mongolicus* (*T. mongolicus*) or *Thymus przewalskii* (*T. przewalskii*), known as “Dijiao (地椒)” in China, which has been included in the *Chinese Pharmacopoeia* (1977) for the treatment of coughs, headaches, abdominal pains, and diarrhea [1]. It is also used as a folk seasoning in some areas of the country [2]. Though *Thymi herba* has been effective and widely used for years, appropriate quality control (QC) criteria, which aim to ensure their consistency, safety, and efficacy, are still lacking. Hence, it is imperative to conduct a suitable QC assessment of *Thymi herba* for the benefit of consumers.

Typically, the QC of herbal medicines is carried out by measuring the chemical markers present in the medicines. Ideally, chemical markers should be unique components that contribute to the therapeutic effects of a herbal medicine [3,4]. Pharmacological studies show that *Thymus* essential oil is the main active component of *Thymi herba*, which has anti-inflammatory [5,6,7], antioxidant [8], antitumor [9], and antithrombotic [10,11] activities.

Thymol (2-isopropyl-5-methylphenol) and carvacrol (5-isopropyl-2-methylphenol) are two main components of *Thymus* essential oil [12,13,14,15,16,17,18]. Thus, for *Thymi herba*, thymol (Thl) and carvacrol (Cal) are considered to be two of the most effective active components (Figure 1). In addition, Thl and Cal have been used in the assay criteria for *Thymus vulgaris* L. and *Thymus zygis* L. in the *European Pharmacopoeia* and the *British Pharmacopoeia* [19]. According to these previous reports, Thl and Cal might be suitable chemical markers for the QC of *Thymi herba*.

Because Thl and Cal are positional isomers, it is usually difficult to completely separate them. So far, only a few methods have been developed for the analysis and quantification of Thl and Cal, such as gas chromatography (GC) and high-performance liquid chromatography (HPLC). The gas chromatography (GC) method (capillary column, column temperature programmed (40 to 220 °C), detector temperature of 230 °C, analysis time 45 min) has been established for the determination of Thl and Cal in *Thymus vulgaris* L. and *Thymus zygis* L. essential oil in the *European Pharmacopoeia* (*European Pharmacopoeia*, 2014, 9, 1538–1560). Apart from this, Pei X. et al. also developed a GC system (capillary column, column temperature programmed (50 to 210 °C), detector temperature of 230 °C, analysis time 25 min) to determine the content of Thl and Cal in *T. mongolicus* (content determination of thymol and carvacrol from *Herba Thyma* by GC. *Chin. J. Exp. Tradit. Med. Form.* 2013, 19, 132–134). Additionally, Ji L. et al. used a high-performance liquid chromatography (HPLC) method (Inertsil ODS-3 column, methanol–water–acetic acid (60:40:2) as mobile phase, analysis time 35 min) to determine the content of Thl and Cal in *Mosla chinensis* (determination of carvacrol and thymol in *Mosla chinensis* by HPLC. *Chin. J. Chin. Mater. Med*. 2004, 29, 8–10). Each method has its unique virtues and drawbacks. GC possesses high selectivity and fine sensitivity [4], but requires extraction of volatile oil [19], operates at high temperatures (40–220 °C), and can require up to 15 min to separate the components [20]. HPLC does not require the extraction of volatile oil, but requires large quantities of organic solvents or chemicals to be consumed, has difficulty in completely separating Thl and Cal, and can require up to 28–31 min to separate the components [21]. Therefore, it is imperative to develop a rapid, reliable, and environmentally friendly method for the QC of *Thymi herba* to ensure its appropriate use for therapeutic purposes.

Ultraperformance convergence chromatography (UPC^2^) has been commercialized by Waters since 2012 [22]. As one of the latest kinds of chromatography technologies, it integrates supercritical fluid chromatography (SFC) and ultraperformance liquid chromatography (UPLC) technologies, which results in many remarkable advantages, including reduced run time, lower solvent consumption, reliability, high resolution, and sensitivity, all of which make its application to routine analysis very attractive. More importantly, UPC^2^ is an environmentally friendly analytical technique that employs dramatically reduced quantities of organic solvents compared to conventional chromatographic methods [23,24]. Nowadays, UPC^2^ technology has been widely utilized in pharmaceutical analysis, and could serve as an alternative or complementary approach alongside HPLC and GC [25,26,27]. To the best of our knowledge, no studies have been reported on the use of UPC^2^ for analyzing Thl and Cal in *Thymi herba* as QC markers.

In this study, a rapid, reliable, and environmentally friendly UPC^2^ method for the simultaneous separation and determination of Thl and Cal in *Thymi herba* was developed and validated for the first time. Using this method, the Thl and Cal contents of 16 samples of *T. mongolicus* and four samples of *T. przewalskii* were analyzed. It is hoped that this study will serve as a guideline for the QC of *Thymi herba*.

## 2. Materials and Methods

### 2.1. Chemicals and Reagents

Reference compounds of Thl and Cal were purchased from the National Institutes for Food and Drug Control (Beijing, China). HPLC-grade methanol and ethanol solvents were obtained from Fisher Scientific Co. (Pittsburgh, PA, USA), CO_2_ (99.999% purity) came from Xenon Heyu Gas Technology Co., Ltd. (Beijing, China), and ultrapure water (18.2 MΩ resistivity) was prepared with a Millipore ultragenetic polishing system (Millipore, Bedford, MA, USA). All solvents were filtered through 0.22 μm filters before use.

*Thymi herba* samples were collected when the flowers were in full bloom in summer and autumn (about June to August). A total of 20 *Thymi herba* samples were purchased/collected from different geographical areas (samples T-1 to T-10 were purchased from different medicinal materials markets on different dates, and samples T-11 to T-20 were collected from different producing areas in June and dried naturally in the shade) (Table 1). All samples were identified as *T. mongolicus* or *T. przewalskii* by Prof. Chunsheng Liu of Beijing University of Chinese Medicine. The voucher specimens (No. ZYS2017T01-T20) have been deposited in the Institute of Chinese Materia Medica, China Academy of Chinese Medical Sciences, Beijing. The samples were deposited in a cool, dry place.

### 2.2. Standard Preparation and Calibration

Stock standard solutions were prepared by dissolving the commercially obtained standards (1.68 mg Thl and 2.01 mg Cal) in methanol (10 mL). Six calibrated standard solutions were prepared using increasing concentrations of the stock solutions and serially diluting them with methanol. A linear correlation between the peak areas of the chromatogram and the concentrations of the components was determined. The linear regression equation obtained was used for quantification of Thl and Cal in selected *Thymi herba* samples.

### 2.3. Sample Preparation

A quantity (3000 g) of the powdered herbal drug was weighed into a 125 mL glass vial with a screw cap. Acetone (Sinopharm Chemical Reagent Co., Ltd, Shanghai, China) (60 mL) was added and the solution was extracted for 40 min using ultrasound. The suspension was then centrifuged for 5 min and the supernatant collected. A quantity (40 mL) of the supernatant was transferred to a flask and evaporated to dryness using a rotary evaporator (at 30 °C and 27 mbar). Methanol (5 mL) was then added to the residue, which was dissolved with the aid of ultrasound for approximately 5 min. The supernatant was filtered through membrane glass fiber filters (0.22 μm, Millipore membrane, Bedford, MA, USA) prior to UPC^2^ (Waters, Milford, MA,USA) analysis.

### 2.4. Apparatus and Separation Conditions

Compounds of interest were analyzed using a Waters ACQUITY^®^ UPC^2TM^ system (Waters, Milford, MA, USA) equipped with a binary solvent delivery pump, an autosampler, a column oven, a back-pressure regulator, and a diode-array detector. The quantitative analysis was performed using a Trefoil^TM^ CEL1 column (2.1 × 150 mm, 2.5 μm, Waters, Milford, MA, USA). The elution gradient of methanol (B) in CO_2_ (A) was used as follows—1% B (initial), 1%–3% B (0–1.5 min), 3%–3% B (1.5–2.5 min), 3%–15% B (2.5–10 min). The automated back pressure regulator (ABPR) was set at 1700 psi. The flow rate was kept at 1.0 mL/min, and the column and autosampler temperatures were set at 30 and 15 °C, respectively. The detection wavelength was set at 274 nm, and the injection volume was 1.0 μL. Data processing was performed using Empower 3 software (Waters, Milford, MA, USA).

### 2.5. Method Validation

The proposed method was validated according to National Medical Products Administration guidelines for the validation of analytical methods for pharmaceutical quality standards, with respect to linearity, limit of detection (LOD) and limit of quantification (LOQ), and for precision and accuracy [28].

Linearity was assessed by plotting the peak area versus concentration of each isomer. The LOD and LOQ were determined as those concentrations where the ratios of the peak heights of interest to the baseline noise were 3 and 10, respectively.

The precision of the analyses for Thl and Cal was determined by three consecutive interday precision measurements and six consecutive intraday precision measurements. The relative standard deviation (RSD) was calculated from the standard deviation (SD) as:SD/mean × 100(1)

The stability of each sample was tested at room temperature (25 °C) and analyzed at 0, 3, 6, 9, 12, 24, 48, and 72 h. Recovery tests were performed by spiking known amounts of the sample to three concentration levels (high, medium, and low) of the standard solution. Recovery was expressed as:(starting concentration-added concentration)/spiked concentration × 100(2)

## 3. Results and Discussion

### 3.1. Optimization of UPC^2^ Conditions

In order to separate Thl and Cal from *T. mongolicus* and *T. przewalskii*, the chromatographic conditions were optimized to achieve good resolution within a reasonable analysis time. First, different columns purchased from Waters were investigated: Waters ACQUITY^®^ UPC^2^™ Viridis BEH (3.0 × 100 mm, 1.7 μm, Waters, Dublin, UK), Trefoil CEL1 (2.1 × 150 mm, 2.5 μm, Waters, Milford, MA, USA), Trefoil CEL2 (2.1 × 150 mm, 2.5 μm, Waters, Milford, MA, USA), Trefoil AMY1 (2.1 × 150 mm, 2.5 μm, Waters, Milford, MA, USA), Torus DEA (2.1 × 100 mm, 1.7 μm, Waters, Dublin, UK), Torus Diol (2.1 × 100 mm, 1.7 μm, Waters, Dublin, UK), Torus 2-PIC (2.1 × 100 mm, 1.7 μm, Waters, Dublin, UK), and Torus 1-AA (2.1 × 100 mm, 1.7 μm, Waters, Dublin, UK) columns. As can be observed, the Viridis BEH and Torus 1-AA columns could not separate Thl and Cal at all. The Trefoil CEL2, Torus DEA, Torus Diol, and Torus 2-PIC columns detected Thl and Cal, but Thl and Cal peaks were not well separated. Only the Trefoil CEL1 and Trefoil AMY1 columns detected Thl and Cal peaks that were well separated. Notably, the Trefoil CEL1 column led to shorter analysis time and better resolution and peak shape (Appendix A). Therefore, the Trefoil CEL1 column was selected as the optimum column for further investigation. Next, in order to improve the separation of Thl and Cal, different solvents for mobile phase B, including methanol, acetonitrile, and ethanol, were investigated. When acetonitrile and ethanol were used as mobile phase B, the peaks of Thl and Cal could not be separated completely. However, the best result was obtained using a mixture of CO_2_ and methanol as mobile phase B with a linear-gradient elution mode. The polarity of methanol is between acetonitrile and ethanol. We speculated that the separation of Thl and Cal may be significantly affected by the polarity of mobile phase B and the polarity of mobile phase B is slightly larger or smaller, which is not conducive to the separation of the two isomers. It is noteworthy that the back pressure directly influences the eluting power of a supercritical fluid by changing the density of supercritical carbon dioxide [26]. Hence, different values of back pressure (1700, 1800, 2000, and 2500 psi) were investigated. With increasing back pressure, the eluting power increased accordingly, resulting in decreased analysis time and resolution. In this work, Thl and Cal were not separated on the back pressure at 1800, 2000, and 2500 psi. Finally, an optimal back pressure of 1700 psi was selected for UPC^2^ analysis. Different column temperatures (30, 35, and 40 °C) and different flow rates (0.7, 1.0, and 1.2 mL/min) were then investigated. The test results showed that the column temperature and flow rates had no significant effect on the resolution of Thl and Cal. However, considering that the Trefoil CEL1 column itself can withstand temperatures not exceeding 40 °C, a lower temperature should be selected without affecting the separation effect. While a flow rate of greater than or equal to 1.2 mL/min resulted in high column pressure, the flow rate was set to 1.0 mL/min, which afforded a lower column pressure and higher resolution. Hence, the optimal column temperature and flow rate were chosen as 30 °C and 1.0 mL/min, respectively. It is important to note that the injection volume directly influences component resolution and peak shape. Therefore, different values of the injection volume (0.5, 1.0, and 2.0 μL) were investigated. With increasing injection volume, the peak shape widened accordingly, resulting in decreased resolution. The optimal injection volume was chosen as 1.0 μL.

Finally, the mobile phase gradient elution program was carefully designed. The smaller the proportion of mobile phase B to supercritical CO_2_, the later that Thl and Cal were eluted. When the elution gradient of methanol (B) in CO_2_ (A) was used as follows—1% B (initial), 1%–3% B (0–1.5 min), and 3%–3% B (1.5–2.5 min)—both Thl and Cal were well separated and the resolutions were good (retention expressed by capacity factor: k_1′_ = 3.67, k_2_^′^ = 4.11; selectivity: α = 1.12; resolution: Rs = 2.00). The formula to calculate the capacity factor: k’; selectivity: α; and resolution: Rs were expressed as:k_1_^′^ = (*t*_R1_ − *t*_M_)/*t*_M_(3)
k_2_^′^ = (*t*_R2_ − *t*_M_)/*t*_M_(4)
α = (*t*_R2_ − *t*_M_)/(*t*_R1_ − *t*_M_)(5)
Rs = (*t*_R2_ − *t*_R1_)/[1/2 × (*W*_1_ + *W*_2_)](6)

(*t*_M_, dead time; *t*_R1_, retention time of thymol (Thl); *t*_R2_, retention time of carvacrol (Cal); *W*_1_, peak width of thymol (Thl); *W*_2_, peak width of carvacrol (Cal)).

### 3.2. Method Validation

#### 3.2.1. Linearity and Sensitivity

Standard stock solutions of Thl and Cal in methanol were prepared. Serial dilution was performed to construct standard calibration curves, with Thl concentrations of 168.00, 84.00, 42.00, 21.00, 10.50, and 5.25 ng/L and Cal concentrations of 201.00, 100.50, 50.25, 25.13, 12.56, and 6.28 ng/L. Good linearity for the two isomers was observed (Table 2), which ensured acquisition of reliable data for different types of samples with both low and high Thl and Cal contents in *T. monoglicus* and *T. przewalskii*.

The LOD was determined as the concentration where the ratio between the peak height of interest and baseline noise was three, and the values ranged from 1.31 to 1.57 ng. The LOQ was determined as the concentration where the ratio between the peak height of interest and baseline noise was 10, and this ranged from 2.63 to 3.14 ng. The result suggested that the UPC^2^ method allows detection of small quantities of Thl and Cal in *T. monoglicus* or *T. przewalskii*.

#### 3.2.2. Precision and Accuracy

The precision of the analyses for Thl and Cal was determined from three consecutive interday precision measurements and six consecutive intraday precision measurements. Interday precision was determined from the standard solutions of the two isomers over three consecutive days. Intraday precision was determined from the standard solutions of the two isomers over six consecutive measurements during a single day. The RSDs of interday precision and intraday precision were 0.71% and 1.16% for Thl and 1.50% and 0.39% for Cal, respectively, indicative of good precision (Table 3). The Thl and Cal were proved to be stable in the sample solution over 72 h at room temperature, with the RSDs below 1.86% and 1.90%, respectively.

The accuracy of the method was determined by spiking known amounts of *T. mongolicus* to three concentrations—low, medium, and high—of the standard solution. All procedures were performed in triplicate, and the sample injections were done in duplicate. The recovery rates for the investigated components ranged from 87.35% to 102.89%, and the RSD values were less than 2.40%, demonstrating that the method we have developed is reproducible, with good accuracy (Table 4).

### 3.3. Quantitative Analysis of T. mongolicus and T. przewalskii

The UPC^2^ method developed in this study was applied for the quantitative determination of the two isomers, Thl and Cal, in 16 samples of *T. mongolicus* and four samples of *T. przewalskii*. Figure 2A shows the typical chromatographic profile of standard solutions of Thl and Cal, and the retention times of Thl and Cal were found to be 2.108 min and 2.248 min, respectively. Figure 2B,C presents, respectively, the representative UPC^2^ chromatograms of *T. mongolicus* and *T. przewalskii*, and Table 5 summarizes the results obtained. Thl and Cal from the aerial parts of *T. mongolicus* were detected in the ranges 0.066–0.224‰ and 0.090–0.616‰, respectively, and Thl and Cal from the aerial parts of *T. przewalskii* were detected in the ranges 0.086–0.246‰ and 0.205–0.323‰, respectively.

This result is in good agreement with the GC quantitative results obtained (Table 6) for these two isomers in *T. mongolicus* in the study reported by Pei X. et al. [20]. It shows that the UPC^2^ method is accurate and reliable, and it may substitute conventional methods used to determine Thl and Cal.

Green analytical chemistry (GAC), which focuses on the development of novel analytical methodologies to reduce the environmental impact of traditional analytical methods, is becoming a more important issue for the public and researchers in recent years [29,30]. Tedious time and organic solvents are always needed to separate the two isomers (Thl and Cal) through the existing GC and HPLC methods, owing to their similar characters. In this work, we have established a UPC^2^ method, and the two isomers are separated efficiently in 2.3 min (6.5-fold shorter (2.3 vs. 15 min) than the GC method and 13.0-fold shorter (2.3 vs. 30 min) than the HPLC method). Meanwhile, a cheap, sustainable, and environmentally benign mobile phase consisting of 1%–3% methanol in CO_2_ is employed in our procedure. The organic solvent consumption and total costs are significantly reduced compared with the previous HPLC method.

Additionally, samples of the same species collected from the different producing region also exhibited considerable differences in Thl and Cal contents. For example, T-1, T-4, and T-9 samples of the same species *T. przewalskii* contained different amounts of Thl and Cal. These results possibly reflect the differences in the quality and bioactivity of *T. przewalskii*.

Hence, we propose that our optimized UPC^2^ method would be particularly convenient for the rapid, accurate, environmentally friendly, and sensitive monitoring the contents of Thl and Cal in *Thymi herba* samples.

## 4. Concluding Remarks

In this study, a UPC^2^ method was established for the first time for the simultaneous determination of the two isomers thymol (Thl) and carvacrol (Cal) in *Thymi herba* (Dijiao, 地椒), which have pharmacological activities. The established method was validated by the linearity, reproducibility, recovery, accuracy, and precision of the results—all parameters were found to be satisfactory. This newly established UPC^2^ method will be helpful in the quality assessment of *Thymi herba* and related herbal formulas in future.

## Figures and Tables

**Figure 1 molecules-25-00502-f001:**
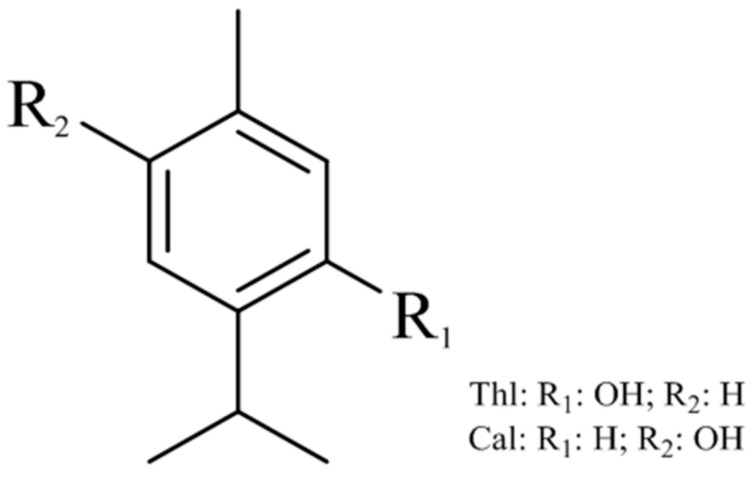
Chemical structures of thymol (Thl) and carvacrol (Cal) (both C_10_H_14_O).

**Figure 2 molecules-25-00502-f002:**
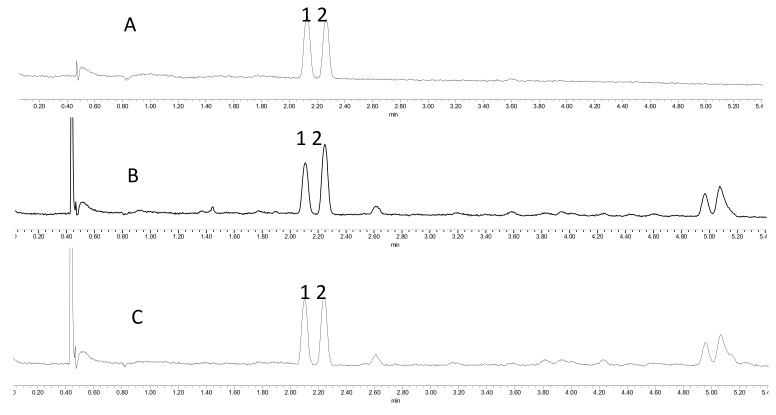
(**A**) UPC^2^ chromatogram of the mixed standard solutions. (**B**) Typical UPC^2^ chromatogram of the *T. mongolicus* extract. (**C**) Typical UPC^2^ chromatogram of the *T. przewalskii* extract. Note: 1, thymol (Thl); 2, carvacrol (Cal).

**Table 1 molecules-25-00502-t001:** Information about *Thymi herba* samples.

Sample	Source	Purchase/Collection date	Species
T-1	Bozhou Sanyitang Pharmaceutical Co., Ltd., Anhui Province	2017.9.5	*T. przewalskii*
T-2	Liupan Shan Town, Ningxia Hui Autonomous Region	2017.9.23	*T. mongolicus*
T-3	Anguo Medicinal Material Market, No. 1, Hebei Province	2017.8.12	*T. mongolicus*
T-4	Kangmei International City of Traditional Chinese Medicine, Anhui Province	2017.10.24	*T. przewalskii*
T-5	Shihezi City, Xinjiang Autonomous Region	2018.1.11	*T. przewalskii*
T-6	Delong County, Ningxia Hui Autonomous Region	2018.1.11	*T. mongolicus*
T-7	Guyuan City, No. 1, Ningxia Hui Autonomous Region	2018.1.18	*T. mongolicus*
T-8	Guyuan City, No. 2, Ningxia Hui Autonomous Region	2018.1.18	*T. mongolicus*
T-9	Haiyuan County, Zhongwei City, Ningxia Hui Autonomous Region	2018.1.18	*T. przewalskii*
T-10	Anguo Medicinal Material Market, No. 2, Hebei Province	2018.3.26	*T. mongolicus*
T-11	Maquan Village, Guyuan City, Ningxia Hui Autonomous Region	2018.6.8	*T. mongolicus*
T-12	Zhangyi Village, Guyuan City, Ningxia Hui Autonomous Region	2018.6.8	*T. mongolicus*
T-13	Songwa Village, Guyuan City, Ningxia Hui Autonomous Region	2018.6.6	*T. mongolicus*
T-14	Pengyang County, Guyuan City, Ningxia Hui Autonomous Region	2018.6.5	*T. mongolicus*
T-15	Dadian Village, Guyuan City, Ningxia Hui Autonomous Region	2018.6.3	*T. mongolicus*
T-16	Xintaozi Village, Guyuan City, Ningxia Hui Autonomous Region	2018.6.6	*T. mongolicus*
T-17	Zhongzhuang Village, Guyuan City, Ningxia Hui Autonomous Region	2018.6.2	*T. mongolicus*
T-18	Yanni Village, Guyuan City, Ningxia Hui Autonomous Region	2018.6.3	*T. mongolicus*
T-19	Tuoxiang Village, Guyuan City, Ningxia Hui Autonomous Region	2018.6.9	*T. mongolicus*
T-20	Zhonghe Village, Guyuan City, Ningxia Hui Autonomous Region	2018.6.10	*T. mongolicus*

**Table 2 molecules-25-00502-t002:** Parameters of the ultraperformance convergence chromatography (UPC^2^) method for determination of Thymol (Thl) and Carvacrol (Cal) in Thymi herba.

Standard	RT (min)	Calibration Curve	*r*	LOD (ng)	LOQ (ng)	Linear Range (ng)
Thymol	2.109	*y* = 602.02*x*	0.9998	1.31	2.63	5.25–168.00
Carvacrol	2.263	*y* = 606.05*x* + 224.73	0.9999	1.57	3.14	6.28–201.00

Calibration curve: *y* = *mx* + *b*, where *y* is the integrated peak area and *x* is the concentration in ng. RT, retention time; LOD, limit of detection; LOQ, limit of quantification.

**Table 3 molecules-25-00502-t003:** Validation of the method for determination of thymol (Thl) and carvacrol (Cal) in *Thymi herba*.

Analyte	Precision (*n* = 6) (RSD, %)		Repeatability (*n* = 6) (RSD, %)	Stability (RSD, %)
	Intraday	Interday		
Thymol	1.16	0.71	1.39	1.86
Carvacrol	0.39	1.50	1.37	1.90

**Table 4 molecules-25-00502-t004:** Recovery of thymol (Thl) and carvacrol (Cal) as determined by the standard addition method (*n* = 6).

Compounds	Sample Weight (g)	Original (mg)	Spiked (mg)	Found (mg)	Recovery (%)	Average recovery (%)	RSD (%)
Thl	1.5041	0.1817	0.0902	0.2745	102.8871	99.1865	2.4016
	1.5056	0.1819	0.0902	0.2707	98.5150		
	1.5000	0.1812	0.0902	0.2732	102.0505		
	1.5006	0.1813	0.1803	0.3618	100.1283		
	1.5043	0.1817	0.1803	0.3634	100.7310		
	1.5019	0.1815	0.1803	0.3572	97.4893		
	1.5098	0.1824	0.2705	0.4451	97.1175		
	1.5001	0.1812	0.2705	0.4460	97.8726		
	1.5019	0.1815	0.2705	0.4408	95.8873		
Cal	1.5041	0.3840	0.1854	0.5513	90.2068	88.7202	1.9644
	1.5056	0.3844	0.1854	0.5465	87.4386		
	1.5000	0.3830	0.1854	0.5458	87.8305		
	1.5006	0.3831	0.3709	0.7235	91.7837		
	1.5043	0.3841	0.3709	0.7211	90.8680		
	1.5019	0.3835	0.3709	0.7072	87.2880		
	1.5098	0.3855	0.5563	0.8714	87.3473		
	1.5001	0.3830	0.5563	0.8699	87.5144		
	1.5019	0.3835	0.5563	0.8742	88.2043		

**Table 5 molecules-25-00502-t005:** Thymol (Thl) and carvacrol (Cal) contents of *T. mongolicus* and *T. przewalskii* (*n* = 2).

Sample	Content (mg/g)	
	Thl	Cal
T-1	0.1643	0.2725
T-2	0.1000	0.1504
T-3	0.1585	0.2192
T-4	0.2458	0.3235
T-5	0.2413	0.2775
T-6	0.1300	0.6159
T-7	0.0959	0.5832
T-8	0.0661	0.5377
T-9	0.0855	0.2051
T-10	0.1956	0.2023
T-11	0.1006	0.5162
T-12	0.1803	0.2128
T-13	0.0695	0.0904
T-14	0.1990	0.2783
T-15	0.1518	0.1439
T-16	0.1577	0.1590
T-17	0.1573	0.1456
T-18	0.0829	0.1680
T-19	0.1203	0.2569
T-20	0.2237	0.2590

**Table 6 molecules-25-00502-t006:** Determination of thymol (Thl) and carvacrol (Cal) contents of *T. mongolicus* by gas chromatography (GC) [20].

Sample	Content (mg/g)	
	Thl	Cal
1	0.255	0.121
2	0.243	0.134
3	0.239	0.118
4	0.216	0.147
5	0.240	0.131
6	0.238	0.122
7	0.120	2.520
8	0.212	1.298

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
