# Peer review of "A New Method for Determination of Thymol and Carvacrol in Thymi herba by Ultraperformance Convergence Chromatography (UPC2)"

_molecules, 2020, doi:10.3390/molecules25030502_

Round 1
Reviewer 1 Report
In the introduction part the authors should introduce exactly how the separation of these isomers it has been done so far
Line 76 –the exact formula of the 2 isomers should be written
In line 85- “A total of 20 air-dried Thymi herba samples were collected from different geographical areas” –this is totally unclear. The air dried samples were collected in a particular date of the year? If so, when?
In line 88- how the samples were deposited?
In table 1, the authors presented the collection date. In all these dates they collected dry samples? How they collect the dry samples in different month?
Line 99-105- The preparation of samples is the authors method?
Line 129- at the formula there is written (2) and (3)
Line 153-156- the authors should detailed at least one example of concentration of methanol in order to delay the peaks
Line 207, Table 5. The authors should add other references in order to compare their results (quantitative data) with other published data (even with other chromatographic condition for isomers separation)
Author Response
Dear Reviewer,
Thanks very much for taking your time to review our manuscript entitled “A new method for determination of thymol and carvacrol in Thymi herba by ultra-performance convergence chromatography (UPC2)” (Manuscript ID: molecules-685974). We really appreciate these precious comments concerning our manuscript, which inspired us to revise and improve our paper. We have studied comments carefully and have made corrections which we hope meet with approval. Accordingly, we have uploaded the file of the revised manuscript with all the changes highlighted by using the track changes mode in MS Word.
Appended to this letter is our point-by-point response to the comments. The comments are listed and our responses are given directly afterward in a different color (blue).
Thanks again!
Yours sincerely,
Xiaoqiang Chang

Reviewer 2 Report
The topic of rewieved manuscript is orginal and innovative. In this work, UPC2 system was used to develop a novel method for the separation of determinatioin of thymol and carvacol in Thymi herba. In my opinion, the present study is interesting and an alternative to conventional methods used to for determining the essential oils. The manuscript is well desined and organized but limitted by some methodical issues appear too great for this to be sent out for peer-review.
In section „Results and discussion” the authors declared that they tested 8 different columns in the present study, but they did not provide a comparison of the results obtained for all columns. Tthey focused only on the column in their opinion best suited for the separation of the analyzed substances.
In my opinion, presenting only validation experiment relevant for quantitative determinations is insufficient. They should provide results describing the efficiency of the chromatographic separation process such as:
Retention expresed by capacity factor [k’]
Selectivity [α]
Resolution [Rs]
it is also possible to place drawings of chromatograms obtained using different columns not in the main body of the manuscript but in suplement file.
The discussion section is written very poor. Try to rewrite the discussion focusing and giving more importance of their own research. Try to discuss how the change of columns and different solvent systems affected the chromatografic optimalization.
Author Response

(The authors gave the same response as above.)

Round 2
Reviewer 2 Report
The manuscript has been improved.
But try to add in methodological section, how did you calculate the capacity factor: k’ Selectivity: α and Resolution
Put the formula
Author Response
Dear Reviewer,
Thank you very much for considering our manuscript entitled “A new method for determination of thymol and carvacrol in Thymi herba by ultra-performance convergence chromatography (UPC2)” (Manuscript ID: molecules-685974) for publication after minor revision. We really appreciate this precious comment concerning our manuscript, which inspired us to revise and improve our paper. We have studied comment carefully and have made corrections which we hope meet with approval.
Accordingly, we have uploaded the file of the revised manuscript with all the changes highlighted (blue) by using the track changes mode in MS Word.
Appended to this letter is our point-by-point response to the comment. The comment is listed and our response is given directly afterward in a different color (blue).
Yours sincerely,
Xiaoqiang Chang
